# Associations between Testing and Game Performance in Ice Hockey: A Scoping Review

**DOI:** 10.3390/sports9090117

**Published:** 2021-08-26

**Authors:** Vincent Huard Pelletier, Julien Glaude-Roy, André-Philipe Daigle, Jean-François Brunelle, Antoine Bissonnette, Jean Lemoyne

**Affiliations:** 1Department of Human Kinetics, Université du Québec à Trois-Rivières, 3351 Boulevard des Forges, Trois-Rivières, QC G9A 5H7, Canada; andre-philipe.daigle@uqtr.ca (A.-P.D.); antoine.bissonnette@uqtr.ca (A.B.); jean.lemoyne@uqtr.ca (J.L.); 2Laboratoire de Recherche sur le Hockey UQTR, Université du Québec à Trois-Rivières, 3351 Boulevard des Forges, Trois-Rivières, QC G9A 5H7, Canada; julien.glaude-roy@uqtr.ca (J.G.-R.); jean.francois.brunelle@uqtr.ca (J.-F.B.); 3Canada Centre de l’Activité Physique et Sportive, Université du Québec à Trois-Rivières, 3351 Boulevard des Forges, Trois-Rivières, QC G9A 5H7, Canada

**Keywords:** functional performance, fitness, sports, athletic development, performance assessment, physical activity

## Abstract

Background: Despite the exhaustive body of literature on the demands of ice hockey, less is known about the relationships between functional performance testing protocols (on ice and off ice) and performance in a game situation. The objective of this review is to provide an overview of these associations. Methods: This review aims to identify on- and off-ice testing currently used in the scientific literature and their possible transfer to game performance as well as identifying research gaps in this field. Results: The 17 selected studies showed that off-ice and on-ice fitness test results can be modestly transferred to the player’s selection as well as global and advanced performance indicators. Conclusion: This review of the literature reinforces the importance of strength and conditioning coaches administering previously validated fitness tests. Regarding the academic research, it is also proposed to use performance markers that are directly related to the players’ on-ice performance to represent more accurately the relationship between the players’ fitness level and their work output. Three research gaps were also identified in relation to targeted populations, choice of performance markers and data measurement methods.

## 1. Introduction

Subsequent to the legendary Summit Series in 1972, ice hockey was recognized as an international sport, which resulted in an invitation to professional players to compete at the Nagano Olympics in 1998. In 2020, over 1.6 million players took to the ice in more than 70 countries [1]. Thanks to this worldwide interest, hockey federations in each nation now look for ways to optimize player development, while aiming to meet the highest standards. This tendency has also generated substantial scholarly interest in the last two decades. For example, a search using the keyword “ice hockey” on the Google Scholar database, filtered for the number of academic publications between 2000 and 2019, revealed a large increase in ice hockey scholar publications: 4000 (2000–2001), 11,000 (2009–2010), and 16,500 (2019–2020).

The seminal works of Montgomery [2], Cox [3], and Bracko [4] provide a comprehensive overview of the intense and multifactorial nature of high-level ice hockey. Sports science publications can in fact be classified into four groups of attributes: physical, technical, tactical, and cognitive-psychological. Morphological characteristics are also believed to play a key role in player selection at the elite level [5]. Even when hockey is played at high intensity, the aerobic system is known to account for about 40% of energy requirements [6]. Anaerobic capacity is also crucial thanks to the high number of short bursts of intensive work during each shift as well as the isometric requirements of the basic skating position, which call for anaerobic processes to provide energy [7]. Furthermore, actions such as absorbing body checks, pushing opponents out of a contested zone and battling for the puck demonstrate how physicality prevails in the nature of the game [7]. Indeed, strength (e.g., maximal force or torque) and power (generating strength at high velocity) are key factors to consider. More attention has recently been given to functional body mobility, which represents a player’s range of motion in the upper and lower limbs. Mobility was shown to be related to better skating techniques and appears to play a role in injury prevention [8].

Technical components refer to specific ice hockey skills. Skating speed [9] and skating agility [10] are two of the most important skills any competitive hockey player must have to excel. Shooting and passing are also integral components of players’ skill set, which is why these are prioritized in the early phases of players’ journey to excellence [11]. The ability to apply tactics, both individual and collective, in offensive and defensive situations is essential in order to achieve the highest level of performance. Individual tactics are situations involving man coverage in one-on-one situations, such as puck protection, efficiency in battles for puck recoveries, and use of the available free spaces. Applying collective tactics or team systems is considered the high point of a player’s performance in the game settings context. A player’s ability to read the game situation and react in harmony with the team’s defensive-offensive system is crucial to his/her use of time over a season [12]. The final category of determinants involves the cognitive and psychological characteristics of the sport. Recently, variables such as decision making, cognitive abilities, grit and resilience have been shown to be key psychological characteristics for determining a player’s long-term success. However, these aspects, their measurements and their transfer to performance in match situations are far less documented than the functional performance tests, so we will focus on the latter in the context of this review.

Generally speaking, hockey organizations (e.g., national teams, federations, professional organizations) aim to identify players with the greatest potential and develop them to meet the highest possible standards. Talent identification is a challenging process that includes several approaches. At the professional level, scouts play a major role in evaluating and ranking players in order to select the best prospects. In scouting, most evaluations derive from the scouts’ subjective judgment and observation [13]. While this approach is based on a prediction of future performance, it has many pitfalls. On one hand, as previously mentioned, approaches to talent detection rely on subjective information processes that offer very little information about a player’s status of development during his progression as an athlete. For this reason, it becomes important to combine subjective assessments of the game skill with objective methods such as functional performance testing and systematic observation of actions in the game.

A second approach to assessing players’ talent or potential and development is based on objective approaches such as testing. Here, players must perform different tests aimed at evaluating their level of ability regarding specific components of performance. Testing or measurements can be conducted in three specific contexts: off ice, on ice, and game settings. Despite this objective method of measuring a player’s level of performance, these approaches have their limitations. Over the years, fitness assessment in ice hockey has been used often and for multiple purposes. In this regard, Quinney et al. [14] demonstrated the relevance and usefulness of off-ice testing by establishing the physical profile of a National Hockey League team across a period of 26 years. Testing can also be used to establish standards for teenage hockey players [15] and elite junior players [16], which allows their development to be observed from a long-term perspective [15]. Off-ice, functional performance tests are also used to assess the impact of training programs [17,18] and develop effective approaches to building training programs [19]. In complement to off-ice functional performance tests, on-ice tests are useful as for determining the physiological demands of hockey. For example, attributes such as aerobic capacity [20], the ability to perform repeated sprints [21,22] and other ice hockey’s physiological demands [23,24] have been observed and defined by scholars.

Despite the usefulness of functional performance tests, their validity for predicting on-ice performance has not yet been established. The relationship between fitness testing and actual in-game performance has been studied in sports such as rugby [25] and soccer [26]. Nightingale [27] questioned the usefulness of ice hockey tests and conducted a review that gives excellent insight into the associations between tests and game performance. Indeed, there seems to be a certain literature gap between the associations involving on and off ice assessments, which limit their role in explaining the game-related performance in ice hockey. For example, Prokop, Reid, and Anderson [28], revealed that off-ice measures are constantly changing during and between seasons, which makes their assessment more difficult to interpret.

## 2. Objectives

Thanks to over fifty years of research, we have a good understanding of the physical demands of hockey and the ways to effectively assess players’ physical attributes. Nevertheless, the question of whether or not these assessments systematically translate to or predict game performance remains unclear. In this regard, we observe a knowledge gap with regard to establishing associations between fitness testing and performance outcomes.

This scoping review has two objectives. First, it aims to identify the studies investigating associations between ice hockey players’ fitness level and player selection (NHL draft, national team selection, player rankings, etc.), global performance indicators (points, games played) and advanced indicators (point share, shot differential, scoring chances). The second objective of this scoping review is to identify research gaps in the literature with regard to fitness indicators and on-ice performances in organized hockey. The goal is to summarize our findings and share them with practitioners and scholars interested in learning more about the associations between fitness testing and game performance.

## 3. Materials and Methods

### 3.1. Identifying the Research Question

In view of our objectives and the absence of a systematic review of the subject, we considered the scoping review the methodology best suited for the study [29]. The review was based on the framework by Levac et al. [30] and thus began with the following research question: to what extent are off-ice and on-ice fitness tests related to players’ performance? In other words, can we assume that a player’s physical attributes assessed in a testing environment are linked to his/her performance in game situations? Answering this question allows physical trainers and coaches to work together to establish testing protocols useful for physical preparation, team selection, and the assessment of players’ potential to perform in competition.

### 3.2. Identifying Relevant Studies

Next, and with the assistance of a university librarian, we searched four different databases (SPORT Discuss, MEDLINE, Pubmed, and Google Scholar Advanced Search) for relevant studies published between 1980 and 2021 using the keywords: “Ice hockey” WITH “predictor-determinant” OR “power” OR “Strength” OR “Endurance” OR “Speed” OR “ Aerobic” OR “Anaerobic” OR “Agility” OR “Fitness” OR “Performance.” We also did a manual search of the grey literature using similar keywords and, finally, another manual research of the lists of references in each identified study.

### 3.3. Study Selection

We extracted references and exported them to Endnote 8 software. To define the inclusion criteria for the articles in our scoping review, the PICO framework was used [31]. Articles on the association between fitness testing (on and off-ice) and hockey performance (player selection, global and advanced indicators) were included, regardless of the nationality, gender or age of the participant. Finally, studies exploring outcomes not directly related to the research question, without peer-review, published before 1990, case studies or non-original research (theses, reviews) were excluded.

### 3.4. Charting the Data

After all relevant studies had been chosen, they were screened by at least two authors to ensure they aligned with the research question. Whenever reviewers disagreed about an article’s relevance, a third reviewer was called on to settle the argument. The data from all studies included were then charted by three authors. V.H.P. extracted data from 90% of the studies and A.P. subsequently extracted data from the remaining 10%. A.P. and J.G.R. checked 20% of V.H.P.’s data extractions for accuracy and then screened the studies using the same process. Any discrepancies were discussed by the authors during their weekly group meeting. Agreement was 90% regarding the studies’ inclusion/exclusion. The three authors (J.G.R., A.P., V.H.P.) discussed the content of the studies on the Excel form to highlight relevant relationships between ice hockey testing and performances.

### 3.5. Collating, Summarizing and Reporting the Results

We analyzed the strength of associations between fitness testing protocols and game performance by taking a deeper look at the statistical outcomes of each study. Because of the different statistical procedures (e.g., correlational, group comparison, descriptive, etc.) used in each study, findings were interpreted in different ways. When possible, we interpreted the magnitude of associations by analyzing the effect sizes from each analysis. For these interpretations, we used Cohen’s [32] guidelines regarding the effect sizes of each result (0.3 = weak, 0.5 = moderate, 0.7 = strong). When additional statistical analyses were performed (see Tables 1 and 2), we used the statistical significance reported in the selected studies. We also considered the proportion of articles that reported favourable associations between tests and performance markers. Additionally, we discussed the mechanisms that could plausibly explain the significant associations relating to the concepts under study.

## 4. Results

### 4.1. Descriptive Statistics

Of the 2364 papers gathered from the search strategy, 259 abstracts and titles (11%) met our inclusion criteria (see Figure 1). After the full-text reading stage of our scoping review, the list of articles was considerably shortened to include 17 peer-reviewed articles retained for further analysis [9,21,33,34,35,36,37,38,39,40,41,42,43,44,45,46,47]. Of these 17 studies, 15 (88%) employed a cross-sectional design and two (12%) were case control studies. Seven studies were published before 2010 (41%), three were published before 2011 and 2015 (18%), while the remaining seven were published after 2016 (see Table 2). All but one study included registered male hockey players (94%). Regarding the leagues of the participants, five studies focused on the collegiate level (29%), five on the junior level (29%), five on various adult elite levels (29%), and two on minor hockey (12%).

### 4.2. Type of Testing: Off-Ice and On-Ice Protocols

Seven studies (41%) combined both on-ice and off-ice tests in their designs [21,33,34,35,36,37,38], eight (47%) included off-ice testing only [9,39,40,41,42,43,44,45], and two study (12%) [46,47] on-ice testing only. In off-ice fitness testing (Table 1), resistance-based tests (e.g., musculoskeletal assessments) predominated and were used in 7 out of the 15 studies [9,21,33,42,43,44,45,46] (46%). Twelves studies (80%) used plyometrics, jump or sprint tests [9,21,33,36,37,38,39,42,43,44,47]. Anaerobic testing was reported in nine (60%) studies [33,34,35,36,37,38,41,45,46], and the Wingate test was the most frequently used. Aerobic tests were less frequent and used mainly with the treadmill, ergocycle and Léger 20 m shuttle tests [21,34,35,36,40,45,47].

For on-ice testing (Table 2), all studies used skating tests assessing speed and acceleration, and a third of them used repeated sprints protocols. The cornering S test, which refers to skating agility, was used in two studies [9,38,39,40,41,42,43,44,45,46]. The other skating tests focused mainly on skating speed and were reported in the remaining studies.

### 4.3. Performance Markers in Game Situations

The three main categories of performance markers used in the selected studies refer to players’ global performance, the metrics associated with specific performance, and team selection, talent identification and/or draft status. For global performance, statistics regarding the number of games played, ice time (playing), and total points (goals + assists) were those reported in three studies (18%) [33,39,42]. Advanced or more game-related metrics were also used in six studies (35%) [21,36,37,41,43,47]. Finally, other metrics related to talent identification were used in nine studies (53%) [9,34,35,38,40,42,44,45,46].

### 4.4. Strength of Associations between Testing and Performance Markers

#### 4.4.1. Off-Ice Testing and Performance Markers

Table 1 presents an overview of studies connecting off-ice test results and hockey performance. Of the seven studies that include resistance-based tests (44%), six report a positive relationship. All three studies using correlations [33,42,46] show a moderate relationship. One study based on group comparison uncovers significant differences in resistance-based test results between players of different calibers of play [9]. Tarter’s paper using ANOVA concludes that 35% of NHL draft selection could be explained with an aggregate score of upper body strength [43], while Vescovi’s study reports no significant effect of fitness test performance on draft selection [44]. Of the 11 studies measuring relationships between muscular power (jumps and sprints) and ice hockey performance, three observe moderate relationships [33,39,40], one indicates a weak association [42], and two find no associations [36,38]. Three studies using group comparison find a significant difference between elite and non-elite players [9,37,45] regarding the muscular power test.

#### 4.4.2. On-Ice Testing and Performance Markers

Table 2 presents studies that estimate the relationships between on-ice testing and hockey performance. Of the six studies (38%) using non-repetitive on-ice time-based tests [34,35,37,38,46,47], three report no association with performance markers [35,38,47], and the three others using group comparison as their statistical analysis find an association based on test results [34,37,46]. Studies by Roczniok (2016) [34] and Bracko (1998) [46] report statistical differences between elite and non-elite players for three of their four on-ice tests. Vigh-Larsen and colleagues (2019) [37] also found significant differences between elite and non-elite players, but with a large effect size. A paper by Roczniok (2013) [35] found that fitness test performance has no significant effect on team selection. Of the six studies (38%) using repetitive on-ice time-based tests [21,34,35,36,37,43], one revealed a moderate relationship [36] and the other reports mixed results [33]. Two other studies based on group comparison observed a significant difference between elite and non-elite players regarding most of the repeated sprint measurements [34,37].

## 5. Discussion

Despite the abundant scientific literature in the ice hockey domain, we have to recognize that most studies aimed at establishing physiological demands of ice hockey [2,3,4,5] and contributed to identifying approaches related to talent identification. This review highlights the scarcity of research regarding on-ice and off-ice fitness tests and their associations with game performance markers. In this regard, this article provided is in line with Nightingale’s paper [27] and add justification for the usefulness of these tests, especially when the objective is to predict players’ performance in real competition settings. With the modest amount of academic literature on this topic, we believe that the inclusion of gameplay performance markers as correlates of functional performance tests is in its early stages of development in ice hockey.

### 5.1. Testing Protocols and Their Associations with Performance

The selected papers showed that the physical attributes that are the more frequently used for off-ice testing are muscular strength, lower limbs’ power and anaerobic capacity. Such attributes are congruent with ice hockey’s physiological demands and refers to key fitness components that might be linked with excellence in ice hockey [4,5]. Nevertheless, we can confirm that the physical attributes tested are generally the same from one study to another. However, our observations suggest that despite the diversity of off-ice fitness tests, no standardized approaches have been used to establish relationships with players’ performance, either in talent identification procedures or in the analysis of game performance. In summary, we can assume that prior studies have helped to identify the tasks of ice hockey players, which allowed the following studies to develop valid and reliable methods to test players’ abilities to perform these tasks [2,3,4,5]. However, as Vescovi stipulated, off-ice tests such as those of the NHL are still modest predictors of future performance. Indeed, more research needs to be conducted regarding the predictive validity of off-ice testing protocols in elite hockey [44].

For on-ice measurements, we observed less variability in the choice of testing protocols. Interestingly, skating speed [2,48], agility [10] and ability to repeat shifts [21,22] are the most frequently used, which corresponds to specific ice hockey demands. Furthermore, skating ability is known as a key ingredient to players’ performance at every stage of development. Despite the specificity of such characteristics, the associations with game performance fluctuate from relatively large (for talent identification) to moderate (game markers). Even if their links with game performance makers are less convincing, we think that this can be explained by the fact that the chosen metrics (e.g., performance markers) in past studies might not have been specifically specific to attributes that were assessed. A good example is the plus–minus differential [49], which reflects a player’s performance without being directly related to a specific ability such as speed, agility, or endurance. From this perspective, we think that researchers should pay more attention to choosing indicators that are related to the specific attributes measured. For example, muscular power could be matched with a player’s success in puck battles and body checks. In addition, we think that variables such as the type of zone entries, back checking and races for loose pucks could be potential correlates of speed and agility. Some promising results were observed in recent research regarding soccer [50], which shows the potential for future research designs. We also think that on-ice protocols should introduce some “specific” ice hockey skills (e.g., shooting, stickhandling, etc.). Protocols such as the one used in Martini’s [22] study is a practical example that allows for measurement of specific hockey players’ skills. Including specific ice hockey skills in testing protocols would, therefore, allow for a more comprehensive and relevant overview of a player’s potential performances in game situations.

### 5.2. Objective 2: Research Gaps and Future Research

The second objective of this scoping review was to identify key research gaps in the area of functional performance testing in ice hockey. The first research gap we have identified is in regard to the selected populations. Most of the selected studies refer to junior, college-varsity and national team players, and very few studies examined female and youth (U18 and younger) hockey. Even if some recent publications focus on female hockey [51,52,53], the importance of female hockey game performance analysis is still modest, and we think that the recent growth in popularity of this discipline will lead to more scholarly publications. This review also showed that little attention was paid to youth hockey players and those at earlier stages of sports development. We believe that monitoring player’s performance at this stage is still relevant, because it allows for a deeper understanding of athletic development and factors that could explain how young athletes are evolving during their early athletic “career”. As suggested by Krajňak [54], puberty is a period in which we observe major differences in physical characteristics such as power, strength and speed. Therefore, we think that it could be relevant to take account of performance markers that could be related to performance at adolescence, for example.

A second research gap in the domain of performance assessment in ice hockey is the choice of performance markers, and the methods used to collect such indicators. The recent emergence of hockey advanced analytics and the different new categories of performance indicators is a promising area of development because they allow researchers to test multiple hypotheses in relation to gameplay performance and see how physical, technical, and psychological attributes can determine how players perform in specific aspects of the game. From this perspective, we think that quantitative game context indicators that can catch specific and measurable attributes would be helpful for authors who aim to explain ice hockey performance. As a third research gap, we think that the methods of data collection need to go further by taking account of more objective indicators instead of considering indicators that are simpler (like draft rankings), which might not be related to the players’ attributes. In this regard, some validation (e.g., construct, concurrent validity) studies are needed to make sure that the objective, advanced indicators are congruent with the attributes under study.

A final research gap concerns data collection methods. Many of the reported papers used systematic observation, box scores analyses, and data collected from video observations. Even if most of these methods are valid, they have some limitations regarding the quantity of information that they provide for researchers. Therefore, the recent emergence of artificial intelligence-based systems or advanced analytics platforms (e.g., InStat, SportlogIQ, etc.) could be a solution for such methodological shortcomings by offering the opportunity to create and analyze large and deep datasets. In addition, we think that using these technologies involving global and local positioning systems (GPS-LPS) that were recently used in the exercise physiology domain [55,56,57] would provide quality data to fulfill this research gap, because it would allow for the measurement of associations between players’ fitness (tested before), on-ice workload during matches, and their performance in specific situations. In fact, sports analytics may be useful for shedding light on the associations between players’ physical attributes (tested in different ways) and their on-ice performance (in different categories of metrics). Linking objective evaluations of successful actions in a game situation to fitness assessment also highlights the importance of making room for the development of other ice-hockey-specific skills in training (e.g., hand–eye coordination, decision making and other perceptive-cognitive skills).

### 5.3. Strengths, Limitations, and Conclusion

This scoping review provided an overview of the current state of research regarding ice hockey performance assessment. The main contribution of this study was to observe the relevance of multiple testing approaches when it comes to the athlete’s follow-up, team selection, talent evaluation, and game performance analyses. In this regard, the present review provides some justification for the usefulness of a different testing approach to establish standards and norms in different playing levels. In a practical way, this review provides interesting insights for coaching staffs, strength conditioning coaches, and professionals involved in talent identification (e.g., scouts, program directors, etc.). For coaching staff and scouts, having a better understanding of the association between their players’ physical attributes profile and the potential outcome during competition might guide them to adjust their rosters for specific events. For strength and conditioning coaches, knowing specifically which functional performance indicators to develop in relation to specific game situations is helpful in designing efficient training programs at different periods of the year.

Despite its contribution, this review also has its limitations. The selected papers came from scholarly journals (written in English language), and some works from the academic domain (like thesis) might not have been selected for this review. In this regard, it is plausible to think that some “unpublished” protocols exist, which might have led to additional observations in this review. We think that during the next decade, new technologies might appear and help assess players physical fitness and on-ice performances. Combining such instrumentation will lead to stronger and more conclusive evidence between functional performance and their outcomes during competition. Another limitation of this paper is that it focuses only on one aspect that can contribute to players’ success on the ice, which is physical fitness. Other technical [58], tactical [59] or psychological [60] aspects would also deserve to be addressed in the future, because they also contribute to game performance. In conclusion, this review showed that most testing protocols are selected according to researchers’ objectives. On the other hand, the choice of performance indicators observed in game situations is vast and could be further explored.

## Figures and Tables

**Figure 1 sports-09-00117-f001:**
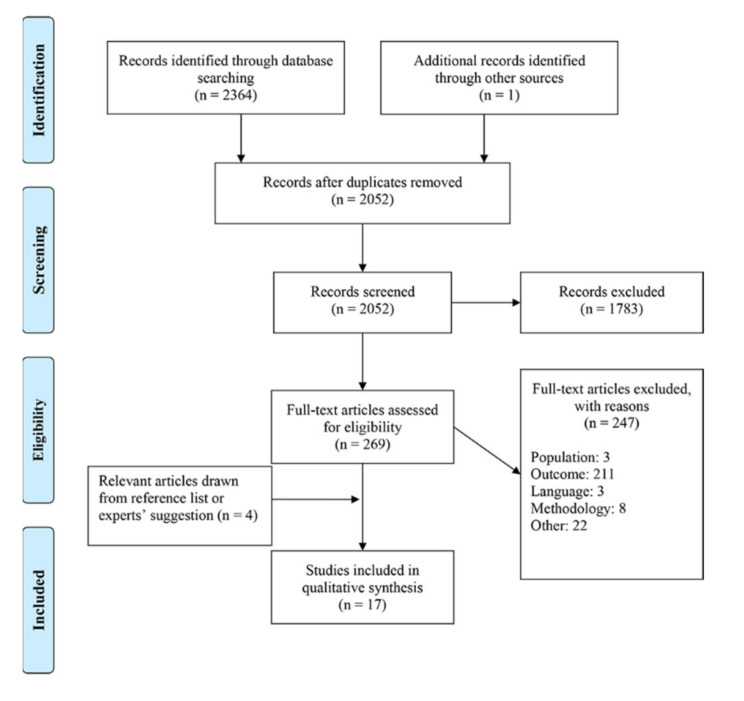
Flow chart.

**Table 1 sports-09-00117-t001:** Associations between off-ice tests and game performance.

References	Population under Study * (*n*)	Off-Ice Tests (Variables)	Performance Markers	Analyses	Strength of Association
Hoff et al., 2005 [9]	Elite (*n* = 18) Elite junior (*n* = 21)	1-RM squat 1-RM bench press CMJ 40 m sprint	Talent identification	T-test	Elite > Junior
Peterson et al., 2015 [21]	Division I (*n* = 24), Junior (*n* = 10) Div. III (*n* = 11)	Squat jump Wingate 30 s Grip strength Skating test-treadmill	Talent identification	ANOVA	DI > D III Junior > D III
Boland et al., 2019 [33]	Female Collegiate (*n* = 20)	Deadlift Max Reps (RM) Bench press 3–5 RM Counter Movement Jump (CMJ) Wingate 30 s	Differential +/− Points Goals Assists Shots on goal	Correlation	None Moderate None Moderate Moderate
Roczniok et al., 2016 [34]	Elite (*n* = 20) Sub elite (*n* = 22)	Wingate 30 s Ramp ergocycle test	Talent identification	T-test	Elite > sub
Roczniok et al., 2013 [35]	Elite (*n* = 22), Sub elite (*n* = 38) Secondary school (*n* = 50)	Wingate 30 s Ramp ergocycle test	Talent identification	ANOVA	Elite > Sec Sub > Sec
Stanula et al., 2018 [36]	National team (*n* = 20)	CMJ Wingate 30 s Incremental test on ergocycle	+/− Differential	Correlation	None
Vigh-Larsen et al., 2019 [37]	Elite (*n* = 164) Sub elite (*n* = 132)	CMJ	Talent identification	T-test	Elite > Sub
Williams & Grau, 2020 [38]	Adolescent high school (*n* = 12)	Standing long jump Single leg lateral jump	Point shares	Correlation	None
Burr et al., 2007 [39]	Junior NHL draft (*n* = 107)	Squat jump CMJ	NHL draft rank	Correlation	Moderate
Delisle-Houde et al., 2018 [40]	Collegiate (*n* = 21)	Standing long jump CMJ Wingate 30 s 20 m shuttle run Léger	Ice time Shift length Shot differential Special teams Playing time	Correlation	Moderate Moderate Moderate Moderate None
Green et al., 2006 [41]	Collegiate (*n* = 29)	Incremental treadmill test	Scoring chances Playing time	Correlation	Moderate Moderate
Kniffin et al., 2017 [42]	Collegiate (*n* = 336)	CMJ Max repetitions bench press	Games played Points	Correlation	Weak Moderate
Tarter et al., 2009 [43]	Junior NHL draft (*n* = 345)	Grip strength Max repetitions bench press Push strength Max repetitions push-ups CMJ Standing long jump Curl ups Wingate 30 s Incremental test on ergometer	Number of NHL games played	Factor analysis	35% chance to make the NHL
Vescovi et al., 2006 [44]	Junior NHL draft (*n* = 250)	Max repetitions push-ups Push strength Pull strength Standing long jump CMJ Sit and reach Wingate 30 s	NHL draft rank	ANOVA MANCOVA	None
Upjohn et al., 2008 [45]	McGill varsity team players and other adult hockey players (*n* = 10)	Treadmill grade Treadmill speed Stride rate Vertical jump Left leg long jump Right leg long jump	Talent identification	T-test	Low-calibre < High-calibre for long jump and treadmill speed
Peyer et al., 2011 [47]	Collegiate (*n* = 24)	Multi-stage treadmill Max repetitions leg press Max repetitions bench press Max repetitions push-ups Max repetitions chin-ups 12 × 110	Differential +/−	Correlation	Moderate

* All populations under study are male, excepted when indicated.

**Table 2 sports-09-00117-t002:** Associations between on-ice tests and game performance.

References	Population under Study * (*n*)	On-Ice Tests (Variables)	Performance Markers	Analyses	Strength of Association
Peterson et al., 2015 [21]	Division I (*n* = 24), Elite junior (*n* = 10) Division III (*n* = 11)	Repeated shift test	Talent identification	ANOVA	DI > DIII Elite jr >D III
Boland et al., 2019 [33]	Female Collegiate players (*n* = 20)	Repeated-Skate-Sprint (RSS) Test	Differential Points Goals Assists Shots on goal	Correlation	None None None Moderate None
Roczniok et al., 2016 [34]	Elite (*n* = 20) Sub elite (*n* = 22)	30 m sprint forward 30 m sprint backwards 6 × 9 m tops 6 × 9 m turns 6 × 30 m	Talent identification	T-test	Elite > Sub
Roczniok et al., 2013 [35]	Elite (*n* = 22), Sub elite (*n* = 38) Secondary school (*n* = 50)	30 m sprint forward 30 m sprint backwards 6 × 9 m tops 6 × 9 m turns 6 × 30 m	Talent identification	ANOVA	None
Stanula et al., 2018 [36]	National team players (*n* = 20)	RSS test	Differential +/−	Correlation	Moderate
Vigh-Larsen et al., 2019 [37]	Elite (*n* = 164) Sub elite (*n* = 132)	5-10-5 33.15 m sprint Yo-Yo Test	Talent identification	T-test	Elite > Sub
Williams & Grau, 2020 [38]	Adolescent high school (*n* = 12)	15 m sprint Cornering S test	Point shares	Correlation	None
Bracko, 1998 [46]	National team Female (*n* = 8) Sub elite female (*n* = 15)	Cornering S test 6.10 m sprint 47.85 m sprint RSS Test	Talent identification	T-test	Nat Team> Sub
Peyer et al., 2011 [47]	Collegiate (*n* = 24)	44 m sprint test Short-lighting test Lap sprint test	Differential	Correlation	None

* All populations under study are male, excepted when indicated.

## Data Availability

Not applicable.

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
