# Peer review of "Associations between Testing and Game Performance in Ice Hockey: A Scoping Review"

_sports, 2021, doi:10.3390/sports9090117_

Round 1

Reviewer 1 Report

The topic of the article is needed for ice-hockey practice, although current research do not provide enough high quality data to check the “Associations between On-Ice Testing and Game Performance“.

The introduction is quite long and do not make straight-forward explanation of review aim. The method section should be more structured and detailed for article selection. Results are acceptable, however their interpretation in discussion is not explained. The discussion is the weakest part of this article and should be re-write to consider (includes only two references), off-ice to on-ice transfer, usefulness of exact testing and relation to on-ice performance measures. The main conclusion and whole results are also not much novel, there is no attractivity and clear and exact concise conclusion.

Specific comments:

Line 21 – 28: There is background is for abstract too long and too misleading. Shorten this part and avoid unproved statements.

Line 27: I thing there are couple of studies dealing with transfer from off-ice to on-ice.

The game situation might be sprint of agility.

Line 33-34: Be more specific for main conclusions, what exactly you can improve?

Line 35 – 36: Some key words are already in the title, use more general ones.

Line 46-48: Your example of increased publication is quite misleading. Why you selecting just one year. Better would be to write such statement generally. Putting such number like 400% increase is not based on standardized calculation.

Line 54: Is any research measured the 70% of aerobic demands during the match?

 Line 66: Reference for agility should not be just general, but also related to ice*hockey measures.

https://www.ncbi.nlm.nih.gov/pmc/articles/PMC6873137/

Line 80: The simple information that this will not be in review is vague. Rather write why you need the conditioning approach to the game performance.

Line 110-120: Please state exact aim of this review the statement “to shad light” on some issue is just vague.

Line 139: The methodology seems to be appropriate. However, it is not clear whether PRISMA statement has been applied. This section should be more structured such as “search strategy” selection criteria etc.

Line 178 – 180: Where in results you actually used the Cohen d? Only for tables?

Line 191 – 196: What is the scope of referring when the article were published. This is not explained in text before.

The discussion is the weakest part of this article and should be re-write to consider (includes only two references), off-ice to on-ice transfer, usefulness of exact testing and relation to on-ice performance measures.

Matthews, Martyn J., Paul Comfort, and Robyn Crebin. "Complex training in ice hockey: the effects of a heavy resisted sprint on subsequent ice-hockey sprint performance." The Journal of Strength & Conditioning Research 24.11 (2010): 2883-2887.

Dæhlin, Torstein E., et al. "Improvement of ice hockey players’ on-ice sprint with combined plyometric and strength training." International journal of sports physiology and performance 12.7 (2017): 893-900.

https://www.ncbi.nlm.nih.gov/pmc/articles/PMC6873137/

Author Response

General comments: The topic of the article is needed for ice-hockey practice, although current research do not provide enough high quality data to check the “Associations between On-Ice Testing and Game Performance“.

The introduction is quite long and do not make straight-forward explanation of review aim. The method section should be more structured and detailed for article selection. Results are acceptable, however their interpretation in discussion is not explained. The discussion is the weakest part of this article and should be re-write to consider (includes only two references), off-ice to on-ice transfer, usefulness of exact testing and relation to on-ice performance measures. The main conclusion and whole results are also not much novel, there is no attractivity and clear and exact concise conclusion.

Comments

Response

Line 21 – 28: There is background is for abstract too long and too misleading. Shorten this part and avoid unproved statements.

We completely agree. Therefore, we removed the first sentence that was not very relevant.

Line 27: I thing there are couple of studies dealing with transfer from off-ice to on-ice.

Yes, that is very true. We changed ‘’No’’ for ‘’Few’’ in the revised version.

The game situation might be sprint of agility.

We did not understand that comment.

Line 33-34: Be more specific for main conclusions, what exactly you can improve?

We tried to be a little more explicit in the new version of the abstract, but the scarcity of the research currently and the variety of the performance indicators it utilized makes generalisation and practical applications difficult.

Line 35 – 36: Some key words are already in the title, use more general ones.

We removed the redundant keyword and added more general ones.

Line 46-48: Your example of increased publication is quite misleading. Why you selecting just one year. Better would be to write such statement generally. Putting such number like 400% increase is not based on standardized calculation.

We selected 3 years (2000, 2010 and 2020) and search in the Google Scholar Database the number of academic publication when the keyword ‘’ice hockey’’ was used just to show the growth in interest. We removed the ‘’400%’’ and replaced it with ‘’large’’.

Line 54: Is any research measured the 70% of aerobic demands during the match ?

Thank you for pointing out this error. We corrected it and changed the associated references.

 Line 66: Reference for agility should not be just general, but also related to ice*hockey measures.

https://www.ncbi.nlm.nih.gov/pmc/articles/PMC6873137/

We replaced the previous reference with the one you proposed. Thank you very much. 

Line 80: The simple information that this will not be in review is vague. Rather write why you need the conditioning approach to the game performance.

We removed this sentence and replaced it with another one to better explain why theses aspects are not part of this review: “These aspects, their measurements and their transfer to performance in match situations are far less documented than the functional performance tests so we will focus on the latter in the context of this review.’’

Line 110-120: Please state exact aim of this review the statement “to shad light” on some issue is just vague.

Yes, this sentence could be quite vague if read alone. However, the subject of the paragraph is clearly indicated in the second sentence of the paragraph (‘’ The relationship between fitness testing and actual in-game performance...’’) and an example is provided in the following sentence (‘’ Prokop, Reid and Anderson [30], for example, reveal that off-ice measures’’).

Line 139: The methodology seems to be appropriate. However, it is not clear whether PRISMA statement has been applied. This section should be more structured such as “search strategy” selection criteria etc.

We agree that this part could have been laid out more clearly. Therefore, we have inserted subheadings to follow Levac's methodological advice. We also replaced ‘’Prisma chart’’ by ‘’Flow chart’’ in the Table 1 title as it is more appropriate.

Line 178 – 180: Where in results you actually used the Cohen d? Only for tables?

Yes, correct.

Line 191 – 196: What is the scope of referring when the article were published. This is not explained in text before.

I'm not sure I understand the question. We mention publication dates to help us describe the sample and allow us to judge the recency of the literature.

The discussion is the weakest part of this article and should be re-write to consider (includes only two references), off-ice to on-ice transfer, usefulness of exact testing and relation to on-ice performance measures.

Matthews, Martyn J., Paul Comfort, and Robyn Crebin. "Complex training in ice hockey: the effects of a heavy resisted sprint on subsequent ice-hockey sprint performance." The Journal of Strength & Conditioning Research 24.11 (2010): 2883-2887.

Dæhlin, Torstein E., et al. "Improvement of ice hockey players’ on-ice sprint with combined plyometric and strength training." International journal of sports physiology and performance 12.7 (2017): 893-900.

https://www.ncbi.nlm.nih.gov/pmc/articles/PMC6873137/

We think that we have improved the discussion in a substantial manner. We adjusted the discussion to establish better relations with the objectives of the article. The discussion has now 6-7 pages and is accompanied by multiple references.

-We added references in the discussion

-We discussed the coherence-validity of testing protocols and associations between tests and performance

-We identified research gaps in this field of research.

-We discussed strengths + limitations+ future perspectives.

-The conclusion was improved.

Reviewer 2 Report

This work presents an interesting scoping review that evaluated the associations between on-ice testing and game performance in ice hockey. The manuscript is well written, clearly structured and the method is well explained. However, I would like to draw the attention of the authors to some points to improve this manuscript. Please find specific comments below.

Title

Based on the guidelines provided by “PRISMA Extension for Scoping Reviews (PRISMA-ScR): Checklist and Explanation”, I propose that the authors include the term “scoping review” in the title of the manuscript.

Abstract

The abstract should be improved and provide a concise description of the methods and findings of the scoping review.

Introduction

Line 115 “Indeed, there seems to be a certain confusion off-ice and on-ice assessments and their role…”. This sentence requires revision.

Line 116: “attempt”. Please use the past tense or the present perfect test when referring to previous studies throughout the manuscript (eg. Line 117 “reveal”, line 248 “find”, line 250 “finds”, lines 252  & 253).

Lines 127 - 137: Even though you state that the scoping review has three objectives (line 132), you describe the first objective in lines 127-131 and the second objective in lines 132-133. Which one is the third objective of the study?  

Materials and methods

Line 158 “(Schardt et al., 2007)” Please assign the appropriate number for this reference.

Figure 1: Even though the records after duplicates were removed were n=2046, the records screened were n=2052. Is this correct?

Results

Lines 240-241 “non-elite players [39, 48] (the study reporting effect size finds a ‘’large difference’’) regarding…” Which study are you referring to?

Discussion

The discussion should be improved and extended. For example, a paragraph or two could be included in the discussion section regarding the strengths and limitations of the existing literature as well as recommendations for future research.

Abbreviations should be defined in parentheses the first time they appear in the manuscript and used consistently thereafter. E.g: Table 2, RSS test.

Author Response

General comments: This work presents an interesting scoping review that evaluated the associations between on-ice testing and game performance in ice hockey. The manuscript is well written, clearly structured and the method is well explained. However, I would like to draw the attention of the authors to some points to improve this manuscript. Please find specific comments below.

Comments

Response

Title: Based on the guidelines provided by “PRISMA Extension for Scoping Reviews (PRISMA-ScR): Checklist and Explanation”, I propose that the authors include the term “scoping review” in the title of the manuscript.

This is a good idea. We thought about this before submitting, and we think this will reflect more precisely the focus of this paper.

Abstract: The abstract should be improved and provide a concise description of the methods and findings of the scoping review.

We tried to be a little more explicit in the new version of the abstract. However, the scarcity of the current research and the variety of the performance indicators it utilized makes generalisation and practical applications difficult.

Line 115 “Indeed, there seems to be a certain confusion off-ice and on-ice assessments and their role…”. This sentence requires revision.

L113-L119: We modified this sentence by establishing more clearly the role of testing (off and on-ice).

Line 116: “attempt”. Please use the past tense or the present perfect test when referring to previous studies throughout the manuscript (eg. Line 117 “reveal”, line 248 “find”, line 250 “finds”, lines 252  & 253).

Thank you for the comment, we made the necessary changes.

Lines 127 - 137: Even though you state that the scoping review has three objectives (line 132), you describe the first objective in lines 127-131 and the second objective in lines 132-133. Which one is the third objective of the study?  

We removed the last sentence and specified that in fact, the review has 2 objectives.

Line 158 “(Schardt et al., 2007)” Please assign the appropriate number for this reference.

We assigned the good reference number and modified the number of those who were reported after.

Figure 1: Even though the records after duplicates were removed were n=2046, the records screened were n=2052. Is this correct?

We have double checked. The good number are now reported.

Lines 240-241 “non-elite players [39, 48] (the study reporting effect size finds a ‘’large difference’’) regarding…” Which study are you referring to?

Thank you for pointing out that error, we removed the parenthesis and its contents, as they had no business being there.

The discussion should be improved and extended. For example, a paragraph or two could be included in the discussion section regarding the strengths and limitations of the existing literature as well as recommendations for future research.

We think that we have improved the discussion in a substantial manner. We adjusted the discussion to establish better relations with the objectives of the article. The discussion has now 6-7 pages and is accompanied by multiple references.

-We added references in the discussion

-We discussed the coherence-validity of testing protocols and associations between tests and performance

- We identified research gaps in this field of research.

-We discussed strengths + limitations+ future perspectives.

-The conclusion was improved.

Abbreviations should be defined in parentheses the first time they appear in the manuscript and used consistently thereafter. E.g: Table 2, RSS test.

We made the corrections for abbreviations that were cited first.

Modifications: see Tables 1 and 2, in which we modified some abbreviations to make it more clear for the reader, and to be in line with the journal’s standards.

Round 2

Reviewer 1 Report

General comments:

The manuscript has been improved in some statements and approaches, however there are still parts which needs major improvement.

The part which should be better explained are the selection criteria: “Articles on the association between testing (on and off-ice) and hockey performance were included, while any studies published before 1990 were excluded. “

Why you did not make search right from 1990 till now?

Please specify why you excluded these studies? One of your aim say “literature with regard to off-ice indicators of on-ice performances in organized hockey.“:

Matthews, Martyn J., Paul Comfort, and Robyn Crebin. "Complex training in ice hockey: the effects of a heavy resisted sprint on subsequent ice-hockey sprint performance." The Journal of Strength & Conditioning Research 24.11 (2010): 2883-2887.

Dæhlin, Torstein E., et al. "Improvement of ice hockey players’ on-ice sprint with combined plyometric and strength training." International journal of sports physiology and performance 12.7 (2017): 893-900.

Pearsall, D.J.; Upjohn, T.; Loh, J. Three-dimensional kinematics of the lower limbs during forward ice hockey skating. Sports Biomech. 2008, 7, 206–221.

You also state: “Finally, studies exploring outcomes not directly related to the research question as along with non-peer reviewed articles, case studies and reviews were excluded.”

Please make explicit selection criteria to be clear.

One of the major limit is lacking the issue of skating technique in ice-hockey youth development and differences between elite and sub-elite level.

Specific comments:

Line 24: The abstract should have conclusion. Now there is a discussion, which do not have discussion form.

Moreover, this part is again without any specific outcome or recommendation.

Stating that “Although the literature is scarce, our study presents a synthesis of the current  state of research in this field, identify research gaps, provide some practical perspectives for athletic trainers and coaches and future insights for researchers”, is very vague. You actually stating this general scope in the aim of the study already.

Authors reply that “generalization and practical applications are difficult.” I agree, but if the authors are not able to make such conclusion, the article do not have a real value.

Generali, stating in abstract that there are some research gaps and some practical perspective mean nothing. There are some Research gaps and practical perspectives out of any research article.

Line 109 -110: The sentence with shading light is redundant. Especially when giving example of anthropometry in next sentence.

Line 116 – 117: You are stating that you observe knowledge gap, which you further confirm by scoping review. In the discussion you should clearly state which “gaps” you potentially resolved by your review.

Line 249 -261: After long introduction, why you are repeating the aim and general attitudes? Delete this paragraph.

322-323: It would be relevant to add also the skating skills in adolescents and youth.

See: https://www.ncbi.nlm.nih.gov/pmc/articles/PMC7552761/

The skating skills are also possible differentiating factor between elite genders and elite-sub-elite men. Try to discuss also this possibility.

Budarick, A.R.; Shell, J.R.; Robbins, S.M.K.; Wu, T.; Renaud, P.J.; Pearsall, D.J. Ice hockey skating sprints: Run to glide mechanics of high calibre male and female athletes. Sports Biomech. 2020, 19, 601–617.

 Pearsall, D.J.; Upjohn, T.; Loh, J. Three-dimensional kinematics of the lower limbs during forward ice hockey skating. Sports Biomech. 2008, 7, 206–221.

Author Response

Comments

Response

The part which should be better explained are the selection criteria: “Articles on the association between testing (on and off-ice) and hockey performance were included, while any studies published before 1990 were excluded. “Why you did not make search right from 1990 till now?

We have paid particular attention to the clarity of our inclusion and exclusion criteria in this new version of the manuscript. However, we do not understand the misunderstanding regarding the dates of publication: we say that we excluded all studies published before 1990. We have therefore included all those published from 1990 to the present. Here is the new ‘’Study selection’’ paragraph:

We extracted references and exported them to Endnote 8 software. To define the inclusion criteria for the articles in our scoping review, the PICO framework was used [31]. Articles on the association between fitness testing (on and off-ice) and hockey performance (player selection, global and advanced indicators) were included, regardless of the nationality, gender or age of the participant. Finally, studies exploring outcomes not directly related to the research question, without peer-review, published before 1990, case studies or non-original research (thesis, reviews) were excluded.

Please specify why you excluded these studies? One of your aim say “literature with regard to off-ice indicators of on-ice performances in organized hockey.

Matthews, Martyn J., Paul Comfort, and Robyn Crebin. "Complex training in ice hockey: the effects of a heavy resisted sprint on subsequent ice-hockey sprint performance." The Journal of Strength & Conditioning Research 24.11 (2010): 2883-2887.

Dæhlin, Torstein E., et al. "Improvement of ice hockey players’ on-ice sprint with combined plyometric and strength training." International journal of sports physiology and performance 12.7 (2017): 893-900.

Pearsall, D.J.; Upjohn, T.; Loh, J. Three-dimensional kinematics of the lower limbs during forward ice hockey skating. Sports Biomech. 2008, 7, 206–221.

We excluded Matthew's study because it does not link a fitness test to on-ice performance. Instead, this study focuses on the benefits of using a loaded sprint on speed development. Here, we test a training method without directly linking the sprint result to on-ice performance (talent identification, global and advanced indicators defined in the ‘’Objectives’’ paragraph).

We rejected the Dæhlin study for the same reason: it compares two approaches to improving speed in players and uses a test to measure this attribute. However, it does not measure how improving speed enhances actual performance on the ice (talent identification, global and advanced indicators).

However, we decided to include the Upjohn study because it links various off-ice tests (long jump, high jump, treadmill) to talent identification. Thank you very much for the suggestion!

One of the major limits is lacking the issue of skating technique in ice-hockey youth development and differences between elite and sub-elite level.

We agree that the technical aspect, as well as the tactical or psychological, would be relevant to address in the future. However, this review focuses only on physical factors and their relation with ice hockey performance.

Line 24: The abstract should have conclusion. Now there is a discussion, which do not have discussion form.

Moreover, this part is again without any specific outcome or recommendation.

Stating that “Although the literature is scarce, our study presents a synthesis of the current state of research in this field, identify research gaps, provide some practical perspectives for athletic trainers and coaches and future insights for researchers”, is very vague. You actually stating this general scope in the aim of the study already.

Authors reply that “generalization and practical applications are difficult.” I agree, but if the authors are not able to make such conclusion, the article do not have a real value.

Generali, stating in abstract that there are some research gaps and some practical perspective mean nothing. There are some Research gaps and practical perspectives out of any research article.

Thank you for your relevant and carefully articulated comments. We think we have improved the abstract by making it less vague and giving readers a foretaste of our results.

We now include 3 research gaps as well as a recommendation for strength and conditioning coaches in the abstract and we removed the sentence starting with ‘’ Although the literature is scarce’’.

Line 109 -110: The sentence with shading light is redundant. Especially when giving example of anthropometry in next sentence.

We agree and have decided to delete the sentence.

Line 116 – 117: You are stating that you observe knowledge gap, which you further confirm by scoping review. In the discussion you should clearly state which “gaps” you potentially resolved by your review.

In the section ''Objective 2: Research gaps and future research''of the discussion, we address the 3 main research gaps that we were able to identify with this review: 1-population studied, 2-performance markers and 3-data collection methods.

Line 249 -261: After long introduction, why you are repeating the aim and general attitudes? Delete this paragraph.

We agree and have decided to delete the paragraph.

 322-323: It would be relevant to add also the skating skills in adolescents and youth. See: https://www.ncbi.nlm.nih.gov/pmc/articles/PMC7552761/

The skating skills are also possible differentiating factor between elite genders and elite-sub-elite men. Try to discuss also this possibility.

Budarick, A.R.; Shell, J.R.; Robbins, S.M.K.; Wu, T.; Renaud, P.J.; Pearsall, D.J. Ice hockey skating sprints: Run to glide mechanics of high calibre male and female athletes. Sports Biomech. 2020, 19, 601–617.

 Pearsall, D.J.; Upjohn, T.; Loh, J. Three-dimensional kinematics of the lower limbs during forward ice hockey skating. Sports Biomech. 2008, 7, 206–221.

As stated earlier, we agree that the technical aspect, as well as the tactical or psychological, would be relevant to address in the future. However, this review focuses only on physical factors and their associations with performance. We added that as one of the study’s limitations (just before the conclusion).

Reviewer 2 Report

The proposed changes have been applied and doubts have been clarified. Nice job.

Author Response

Thank you very much for your constructive comments that helped us improve the manuscript. 

Round 3

Reviewer 1 Report

The authors explained most of my concerns and significantly improved the text and study conclusions. Therefore, I recommend this article to be published.